# Exploratory study of risk factors related to SARS-CoV-2 prevalence in nursing homes in Flanders (Belgium) during the first wave of the COVID-19 pandemic

Heidi Janssens[1,2]*, Stefan Heytens[2], Eline Meyers[3], Brecht Devleesschauwer[4,5], Piet Cools[3©], Tom Geens[1©]

1 Research and Analytics, Liantis, Belgium, 2 Department of Public Health and Primary Care, Faculty of Medicine and Health Sciences, Ghent University, Ghent, Belgium, 3 Department of Diagnostic Sciences, Faculty of Medicine and Health Sciences, Ghent University, Ghent, Belgium, 4 Department of Epidemiology and Public Health, Sciensano, Brussels, Belgium, 5 Department of Translational Physiology, Infectiology and Public Health, Faculty of Veterinary Sciences, Ghent University, Merelbeke, Belgium

© These authors contributed equally to this work.
* heidi.janssens@liantis.be

**Data Availability Statement:** Data are available at https://github.com/HeidiJanssens/SARS-CoV-2-Liantis-study.

## Abstract

In a previous study in Belgian nursing homes (NH) during the first wave of the COVID-19 pandemic, we found a SARS-CoV-2 seroprevalence of 17% with a large variability (0–45%) between NH. The current exploratory study aimed to identify nursing home-specific risk factors for high SARS-CoV-2 seroprevalence. Between October 19th, 2020 and November 13th, 2020, during the second COVID-19 wave in Belgium, capillary blood was collected on dried blood spots from 60 residents and staff in each of the 20 participating NH in Flanders and Brussels. The presence of SARS-CoV-2-specific IgG antibodies was assessed by ELISA. Risk factors were evaluated using a questionnaire, filled in by the director or manager of the NH. Assessed risk factors comprised community-related factors, resident-related factors, management and performance features as well as building-related aspects. The relation between risk factors and seroprevalence was assessed by applying random forest modelling, generalized linear models and Bayesian linear regression. The present analyses showed that the prevalence of residents with dementia, the scarcity of personal protective equipment (surgical masks, FFP2 masks, glasses and face shields), and inadequate PCR test capacity were related to a higher seroprevalence. Generally, our study put forward that the various aspects of infection prevention in NH require more attention and investment. This exploratory study suggests that the ratio of residents with dementia, the availability of test capacity and personal protective equipment may have played a role in the SARS-CoV-2 seroprevalence of NH, after the first wave. It underscores the importance of the availability of PPE and education in infection prevention. Moreover, investments may also yield benefits in the prevention of other respiratory infections (such as influenza).

**Funding:** P.C. received funding from the Special Research Fund of Ghent University (BOF. COV.2020.0010.01). The funder had no role in study design, data collection and analysis, decision to publish or preparation of the manuscript.

**Competing interests:** The authors have declared that no competing interests exist.

## Introduction

In Europe, nursing homes (NH) were disproportionately hit by the COVID-19 pandemic, resulting in estimates of about 50% of the COVID-19 deaths being NH residents at the end of June 2020 [1]. Belgium was one of the countries with the highest number of reported COVID-19 deaths per capita, with NH accounting for about 60% of these deaths [2]. A lot of research has focused on resident related characteristics to explain this particular vulnerability of NH. First, due to older age and multiple underlying diseases [3], this resident population is specifically at risk for more severe and fatal presentation of the infection. Moreover, older persons often present atypical COVID-19 symptoms [3] or remain asymptomatic [4, 5]. This results in a delayed diagnosis, which poses difficulties for early isolation leading to spread among residents [6]. Finally, residents have, due to their physical and cognitive impairments, a lot of contacts with their caregivers, which interferes with the physical distancing rules.

Aggregate numbers of infection rates in NH are higher than in the general population [5], nevertheless, large differences can be noticed between the individual NH, with some of them severely hit, while others are spared [5, 7–9]. In order to understand the differences between these NH, a number of studies, mainly from the United States and Canada, studied risk factors related to the residential care facility and proxy variables for the surrounding regional infection incidence. Risk factors that are commonly (but not consistently) related to introduction of SARS-CoV-2 in the NH, and/or transmission rates and/or higher prevalences in these studies are higher number of beds [7, 10–12], higher incidence of COVID-19 in the region or county [7, 8, 11, 13, 14], larger community population size [8, 10, 13], higher crowding or density of residents within NH [8, 13] and lower NH quality ratings [14]. Generally, a lot of outbreaks in NH have been attributed to inadequate implementation of infection control [15], test strategy, early diagnosis and contact restriction [16].

In Belgium, studies examining the relation between NH specific risk factors and infection prevalence or incidence are limited. Peckeu-Abboud and colleagues [17] investigated NH characteristics associated with SARS-CoV-2 test positivity rate among residents in 695 Flemish NH. They found that the proportion of beds for residents with a high degree of care dependency as well as positivity rate among staff were related to an increase in proportion of residents tested positive during the first wave of the pandemic. No significant relation was found with the other considered risk factors (number of beds, ratio of staff members/residents; age of residents and staff; proportion of asymptomatic cases for both residents and staff; public, private non-profit and private for-profit institutions). However, no information was gathered on infection control procedures (isolation procedures or staff cohorting) and testing strategies.

Therefore, this exploratory study aimed to assess risk factors (including infection control procedures) for high SARS-CoV-2 seroprevalence, as a measure for viral spreading in Flemish NH. Risk factors were categorized in 4 groups, following the approach of Zhu [9]: factors related to (1) the community, (2) the residents characteristics, (3) the NH management and performance, (4) the building of the NH.

## Material and methods

### Study design

The current manuscript is the second report from the overarching SARS-CoV-2 Liantis study. The whole cross-sectional study was designed to (i) assess the seroprevalence in Belgian NH after the first wave [5], and (ii) identify risk factors at both the individual (residents and staff) and NH level for SARS-CoV-2 infection in the first SARS-CoV-2 wave in Belgium (current report). To cover the purpose of both study objectives, sample size calculations demonstrated

that 80 NH, with each 60 participants, across Belgium were needed. To anticipate non-response, 100 NH were initially selected.

The whole study design, selection and recruitment procedure are extensively described elsewhere [5]. Between 19 October and 13 November 2020, the SARS-CoV-2 Liantis study tested NH residents and staff for the presence of SARS-CoV-2 IgG antibodies and collected demographic, behavioural, clinical and NH-specific data through questionnaires conducted in residents, staff and management of the NH.

## Selection and recruitment of the NH

The NH were selected from a database of Liantis, a Belgian external occupational health service. A subset of 210 NH employing at least ten staff members was used. In order to obtain a representative

geographical sample of the Belgian NH, the 100 NH were chosen according to the true proportion of NH per province in Belgium, based on data available in the Crossroads Bank for Enterprises [18].

The end of the first epidemic wave in Belgium was retrospectively defined and set on 22 June 2020 [19]. The start of the second wave was set on 31 August 2020 [19]. Due to the rapid progression of the 2nd wave of COVID-19 in Belgium [20], which initially affected particularly the Walloon region in Belgium, sampling was prematurely stopped, even before any sampling was carried out in Wallonia. Since one of the study goals was to describe NH-specific risk factors that occurred during the first wave, further sampling would bias the relationship between these risk factors and prevalence. In this way, we found 20 NH from the 56 originally selected NH, in the Belgian Flemish and Brussels Capital Regions, willing to participate.

## Selection of the staff and residents

In every included NH, 60 residents and staff were randomly selected using an online tool specifically developed for this study. The proportion of selected residents and staff in each NH reflected the proportion of residents and staff in that NH. In one smaller nursing home, the selection was limited to 45 residents and staff due to limited number of staff. For the staff members, there were no exclusion criteria. For the residents, those living in assisted living facilities were excluded. This was done since important differences between NH and assisted living facilities concerning social interactions and caring approach exists [21], which would have hampered the questionnaire design. For each staff member or resident who refused to participate, an additional randomly selected staff member or resident was invited to participate. This selection procedure resulted in median number of staff participants of 25 (minimum: 19, maximum: 32) and a median number of residents of 35 (minimum: 24, maximum: 41).

## SARS-CoV-2 serology

To assess the presence of SARS-CoV-2-specific IgG antibodies, capillary blood was collected via a finger prick from each participant on protein saver cards and air-dried (i.e., dried blood spots, DBS) as previously described [22]. DBS were sent to the Laboratory Bacteriology Research (LBR, Department Diagnostic Sciences, Faculty of Medicine and Health Sciences, Ghent University), and analyzed for the presence of anti-spike (anti-S) IgG antibodies by means of ELISA (EUROIMMUN, PerkinElmer Health Sciences Inc., Lübeck, Germany) [22]. Participants were classified as seronegative or seropositive according to the optical density ratio of the sample, retrieved by ELISA ($<0.8$ or $\geq 0.8$ respectively).

## Questionnaires

A NH-specific questionnaire was designed in LimeSurvey [23]. This questionnaire was designed based on literature review, the COVID-19 prevention guidelines proposed by the authorities and feedback from the working field. The questions asked about the facility itself (such as the characteristics of the building, the infrastructure and the number of beds), the extent to which the facility was affected by the pandemic, staff and resident demographics, frailty of the residents, and prevention measures taken during the different stages in the first wave of the pandemic. The questionnaire was designed and improved iteratively in cooperation with a number of the present co-authors and volunteers from the target population. Questionnaires were available in Dutch and French and after receiving a unique code, the NH-manager or director got access to the digital version [23]. The English translated version of the NH questionnaire is found in S1 File.

## Statistical analysis

LimeSurvey online questionnaire responses were exported as CSV files and imported into R data frames. Statistical analyses were performed using R 4.1.3 [24].

Prevalences (and 95% confidence intervals) were reported as frequencies of positive SARS-CoV-2 antibody tests proportional to the total sample size, using a one sample proportion test with continuity correction.

Selection of risk factors was conducted in a number of steps. In a first step, only the questions asking for potential risk factors were selected from the NH questionnaire: this implies that questions assessing the consequences of the pandemic (such as sickness absence of the staff) or the severity in which a NH was affected (such as number of deaths) were not retained. The distribution of the answers was checked. From this first exploratory step, it became clear that questions with a specific calendar date during the first pandemic wave as answer (for instance: "From when did you begin applying these insulation measures to residents (approximately)?" Enter a date: . . .) were not useful, nor without, nor with categorization. The answers to questions concerning behavior (such as wearing masks) may be less reliable and were therefore not retained.

In a second step, the remaining number of risk factors was reduced. One strategy was choosing one risk factor for questions which were asking related topics. The answers on the questions 'What is the year of construction of your facility's (main) building?' and 'Has the building already been renovated?' assessed related aspects. Therefore, only the construction year was retained as a risk factor. Also the seven questions assessing the feeling if the staff was trained to use the personal protective equipment (PPE) were evaluating associated issues. The answers on these questions were summed up to retain one overall score. For clusters of questions on risk factors assessing related topics, Principal Component Analysis with varimax rotation was applied to identify and retain less clear underlying risk factors (for instance: the seven sub-questions assessing the availability of PPE). In case such an underlying risk factor could be found, information from the original questions was combined by summing the questions up, in order to reduce the number of risk factors. In S1 Table, the selected variables/risk factors, with referral to the respective questions on which they are based, are summarized. Urbanisation, type of organisation, PCR-capacity, room cleaning and year of construction of the building were used in an unchanged form as they were asked in the questionnaire (see S1 Table: [a] marked questions). Three ratio variables were calculated by normalization: the number of females, older residents and residents with dementia through division with the total number of residents. The questions assessing the availability of the PPE consist of seven sub-questions (surgical masks, FFP2-filtermasks, disposable gloves, safety glasses, aprons, face-shields and disinfectant alcohol), with four answer options varying between definitely insufficient to

definitely sufficient. After conducting a Principal Component Analysis, the seven features could be reduced to two components (one referring to the masks, goggles and face-shields, and one referring to the aprons, gloves and disinfectant alcohol). Thereafter the answers on the individual questions were summed (1 = definitely insufficient, 4 = definitely sufficient), creating two continuous variables reflecting the availability of the PPE. The seven questions assessing the feeling that the employees are adequately trained to use the PPE (with four response options; 1 = no, not at all; 2 = rather not: 3 = rather yes; 4 = yes, definitely) are summed up and treated as one continuous variable. Finally, the answers on the two questions regarding the ventilation of rooms (four response options) and common rooms (three response options) were combined into one variable, with four response categories (see S1 Table)

In a third and final step, three different approaches were used to explore the relations between the reduced number of risk factors and seroprevalence at NH level. Random forest modelling, using the randomForests package [25] was applied to examine the relative importance of the risk factors (using both the incnodpurity & %mse approach). Generalized linear models for a binary outcome with a logit link, using the glm function from the stats package [24], were applied to assess the univariable association between each risk factor and seroprevalence. Since the number of cases for a multivariable model in a generalized linear model approach is too low, Bayesian generalized linear regression, using the stan glm function from the rstanarm package, [26] was applied to estimate multivariable relations between risk factors and seroprevalence. Default prior distributions settings were used for coefficients and intercept and four Markov chains were used with 2000 iterations each (1000 for warmup, 1000 for sampling).

## Ethical considerations

Ethical approval was obtained by the Ethical Committee of the Ghent University Hospital (reference number BC-07665). The NH management informed residents and their families, and staff on the study objectives and sampling procedures. Residents (and/or family of residents) and staff who agreed to participate in the study signed an informed consent form. A confidential counsellor (a family member or a nurse after approval by the family) signed for participants who were incapable to sign the consent form, such as residents with dementia. The authors did not have access to information that could identify individual participants during or after data collection: they only had codes with serology and questionnaire results. Participant lists with names and codes were only available to the director of the NH, acting as a thrusted third party.

## Results

### Seroprevalence

From 478 staff members and 615 residents from the 20 NH (1093 of the 1185 included participants which corresponds to 92.2% of the total study population), a valid serological test result was obtained. Main reasons for this missing information were the absence of the participant at the moment of sample collection or the impossibility to obtain enough blood. The overall seroprevalence was 17.1% (95% Confidence Interval (CI), 14.9–19.5), with 18.9% (95% CI, 15.9–22.2) of the residents and 14.9% (95% CI, 11.9–18.4) of the staff having antibodies. The seroprevalence per NH varied between 0% and 45.0% (see Fig 1).

### Overview of the selected risk factors

In Tables 1 and 2, an overview of the results of the categorical and continuous risk factors is given. Twenty percent of the NH reported that they experienced not enough PCR testing capacity. Respectively 35% and 30% of the NH reported they had not enough FFP2 masks and

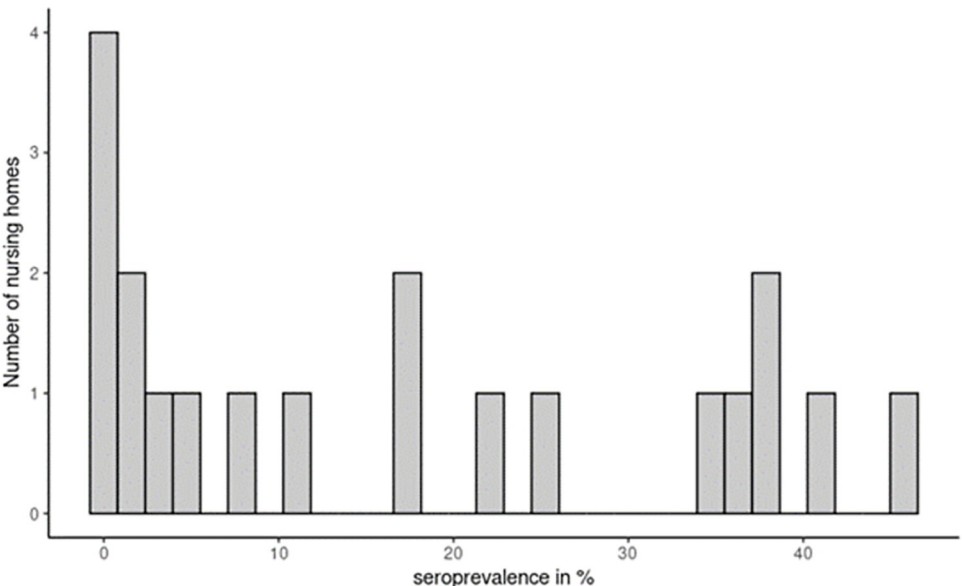

**Fig 1. Graphical presentation of the distribution of seroprevalence (%) across the 20 nursing homes.**

gloves. The percentage of residents with dementia varied between 17 and 50% in the participating NH.

## Simple (univariable) associations between risk factors and seroprevalence (Table 3)

The univariable results with significance level lower than 0.001 show urbanisation is a risk factor for higher seroprevalence. As risk factors among residents, the percentage of residents with dementia was retained.

In general, a lower degree of ventilation was correlated to higher seroprevalence. Regarding NH policy risk factors, the following aspects were correlated to higher seroprevalence: insufficient PCR testing capacity, lower availability of certain PPE (masks, face and eye protection), and feeling less educated to use the PPE.

## Results of the multivariable associations between risk factors and seroprevalence based on Bayesian generalized linear model

The results of the multivariable model using Bayesian generalized linear regression modelling (Table 3) demonstrate that the percentage of residents with dementia, having enough testing capacity with government test rounds/with government tests rounds and own resources and better availability of disinfectant alcohol, gloves and aprons was related to higher seroprevalence in NH.

Better availability of masks (FFP2 and surgical), eye (safety glasses) and face protection (face shields) was related to lower seroprevalence, as was ventilation of rooms (by opening windows) and regular ventilation of the common areas.

## Results of the random forest analysis (Fig 2)

The random forest modeling (based on the Gini impurity index) suggests that the lack of availability of masks, face and eye protection, testing capacity, high percentage of residents with dementia, and not feeling adequately educated were the main risk factors for higher

**Table 1. Categorical risk factors in the twenty included nursing homes (NH).**

| Variable | Categories | Number (%) of NH |
|---|---|---|
| **Type of NH** | | |
| | Public | 5 (25) |
| | Private | 3 (15) |
| | Non profit | 12 (60) |
| **Urbanisation of geographical location of NH** | | |
| | City | 4 (20) |
| | Periphery | 12 (60) |
| | Rural area | 4 (20) |
| **PCR-test capacity** | | |
| | Yes, but only on own resources | 5 (25) |
| | Yes, with the government test rounds | 5 (25) |
| | Yes, with the government test rounds and on own resources | 6 (30) |
| | No | 4 (20) |
| **Evaluation of the availability of (personal) protective equipment (for employees)?** | | |
| Disposable gloves | Definitely insufficient | 1 (5) |
| | Insufficient | 5 (25) |
| | Sufficient | 7 (35) |
| | Definitely sufficient | 7 (35) |
| Surgical mouth masks | Definitely insufficient | 2 (10) |
| | Insufficient | 0 (0) |
| | Sufficient | 13 (62) |
| | Definitely sufficient | 5 (25) |
| FFP2 filter masks | Definitely insufficient | 3 (15) |
| | Insufficient | 4 (20) |
| | Sufficient | 11 (55) |
| | Definitely sufficient | 2 (10) |
| Disposable aprons (long sleeves) | Definitely insufficient | 2 (10) |
| | Insufficient | 3 (15) |
| | Sufficient | 12 (60) |
| | Definitely sufficient | 3 (15) |
| Face-shields | Definitely insufficient | 1 (5) |
| | Insufficient | 3 (15) |
| | Sufficient | 9 (45) |
| | Definitely sufficient | 7 (35) |
| Safety glasses | Definitely insufficient | 2 (10) |
| | Insufficient | 2 (10) |
| | Sufficient | 10 (50) |
| | Definitely sufficient | 6 (30) |
| Hand alcohol gel | Definitely insufficient | 1 (5) |
| | Insufficient | 0 (0) |
| | Sufficient | 12 (60) |
| | Definitely sufficient | 7 (35) |
| **Daily cleaning and disinfecting of rooms with a product active against SARS-CoV-2, from the beginning of the pandemic?** | | |
| | Yes | 11 (55) |
| | No | 9 (45) |

(*Continued*)

**Table 1.** (Continued)

| Variable | Categories | Number (%) of NH |
|---|---|---|
| **Do you feel that staff members were adequately trained to properly use this material?** | | |
| Disposable gloves | Yes, definitely | 10 (50) |
| | Rather yes | 8 (40) |
| | Rather not | 2 (10) |
| | No, not at all | 0 (0) |
| Surgical mouth masks | Yes, definitely | 8 (40) |
| | Rather yes | 9 (45) |
| | Rather not | 2 (10) |
| | No, not at all | 1 (5) |
| FFP2 filter masks | Yes, definitely | 5 (25) |
| | Rather yes | 11 (55) |
| | Rather not | 3 (15) |
| | No, not at all | 1 (5) |
| Disposable aprons (long sleeves) | Yes, definitely | 5 (25) |
| | Rather yes | 9 (45) |
| | Rather not | 6 (30) |
| | No, not at all | 0 (0) |
| Face-shields | Yes, definitely | 8 (40) |
| | Rather yes | 10 (50) |
| | Rather not | 1 (5) |
| | No, not at all | 1 (5) |
| Safety glasses | Yes, definitely | 8 (40) |
| | Rather yes | 11 (55) |
| | Rather not | 1 (5) |
| | No, not at all | 0 (0) |
| Hand alcohol gel | Yes, definitely | 16 (80) |
| | Rather yes | 4 (20) |
| | Rather not | 0 (0) |
| | No, not at all | 0 (0) |
| **Structurally naturally ventilation of rooms at the beginning (first weeks) of the COVID-19 epidemic?** | | |
| | Yes, through opening of windows | 7 (35) |
| | Yes, through opening of doors | 0 (0) |
| | Yes, through opening of windows and doors | 11 (55) |
| | No | 2 (10) |
| **Structurally naturally ventilation of common areas at the beginning (first weeks) of the COVID-19 epidemic?** | | |
| | Yes, structurally and continuous ventilation was provided | 4 (20) |
| | Yes, at regular times (e.g. after increased use of the room) | 14 (70) |
| | No | 2 (10) |

seroprevalence. In an alternative approach using %mse criterion, also the lack of availability of PPE, testing capacity were retained as risk factors with the highest relative importance, followed by the high ratio of female residents (see S1 Fig).

## Discussion

As former research showed a large variation in seroprevalence between NH [5, 27], we hypothesized that factors related to the surrounding community and factors specifically related to the

**Table 2. Continuous risk factors in the twenty included nursing homes (NH).**

| | Median (minimum-maximum) in the 20 NH |
|---|---|
| Ratio females (number of female residents divided by the total number of residents, in percent) | 72% (23%-91%) |
| Ratio older residents (number of residents older than 84 years divided by the total number of residents, in percent) | 67% (8%-87%) |
| Ratio residents with dementia (number of residents with dementia divided by the total number of residents, in percent) | 38% (17%-50%) |
| Construction year of the building | 2000 (1958–2020) |

**Table 3. Overview of the risk factors, with respectively regression coefficient and significance level of the univariable relations with seroprevalence, estimated with generalized linear models (1) and median and 95% credible interval (between squared brackets) of the multivariable relations with seroprevalence, estimated with Bayesian generalized linear regression (2).**

| Risk factor | 1 | 2 |
|---|---|---|
| **Risk factors related to the community** | | |
| Urbanisation (Urbanisation) | | |
| *City* | Ref. | Ref. |
| *Peripheral* | **-0.95**\*\*\* | -1.60 [-6.69, 4.31] |
| *Rural* | **-1.12**\*\*\* | -1.71 [-5.45, 2.38] |
| **Risk factors related to NH residents' characteristics** | | |
| Ratio females (Ratiofemales) | -1.28\*\* | 6.37 [-3.10, 15.33] |
| Ratio older residents (> = 85 years) (Ratio_85_and_older) | -0.81 | 5.44 [-5.61, 17.37] |
| Ratio residents with dementia (Ratio_dementia) | **8.14**\*\*\* | **14.36 [3.32, 27.32]** |
| **Risk factors related to NH management and performance** | | |
| Type of organisation (Organisation_type) | | |
| *Private* | Ref. | Ref. |
| *Public* | 0.94\*\* | 2.06 [-0.42, 4.99] |
| *Non-profit* | 0.20 | 0.69 [-2.05, 3.48] |
| Enough PCR testing capacity, when needed? (PCR_capacity) | | |
| *Yes, but only with own resources* | Ref. | Ref. |
| *Yes, with the government test rounds* | 0.08 | **3.62 [0.04, 7.35]** |
| *Yes, with the government test rounds and own resources* | 0.59\* | **5.61 [0.56, 10.85]** |
| *No* | **1.86**\*\*\* | 1.89 [-1.94, 5.93] |
| Personal protective equipment (masks, face and eye protection) (PPE1) | **-0.87**\*\*\* | **-3.58 [-6.40, -1.42]** |
| Personal protective equipment (gloves, aprons and disinfectant alcohol) (PPE2) | -0.17 | **2.49 [0.58, 4.52]** |
| Room cleaning? (Cleaning) | | |
| *Yes* | Ref. | Ref. |
| *no* | 0.32\* | -1.04 [-3.76, 1.03] |
| Trained to use PPE (Education) | **-0.12**\*\*\* | 0.05 [-0.27, 0.38] |
| **Risk factors related to the building** | | |
| Year of construction of the building (Construction_year) | 0.01\* | 0.04 [-0.01, 0.10] |
| Combined variable related to the degree of natural ventilation (Ventilation) | | |
| *No ventilation of rooms & common rooms* | Ref. | Ref. |
| *Ventilation of rooms (by opening windows) & regular ventilation of common rooms* | **-0.87**\*\*\* | **-5.13 [-8.51, -1.99]** |
| *Ventilation of rooms (by opening doors) & regular ventilation of common rooms* | **-1.04**\*\*\* | -2.62 [-8.47, 3.25] |
| *Ventilation of rooms (by opening windows and doors) & continuous ventilation of common rooms* | -0.24 | -2.50 [-6.89, 1.76] |

Results in bold are significant at 0.001 level (after correction for multiple testing) in univariable model and when zero is not contained in the credibility interval in the Bayesian approach; Ref., Reference category; factors between round brackets refer to the variable name in the Random Forest modeling in Fig 2 and S1 Fig

\*\*\*p- value <0.001

\*\* p-value <0.01

\*p-value <0.05

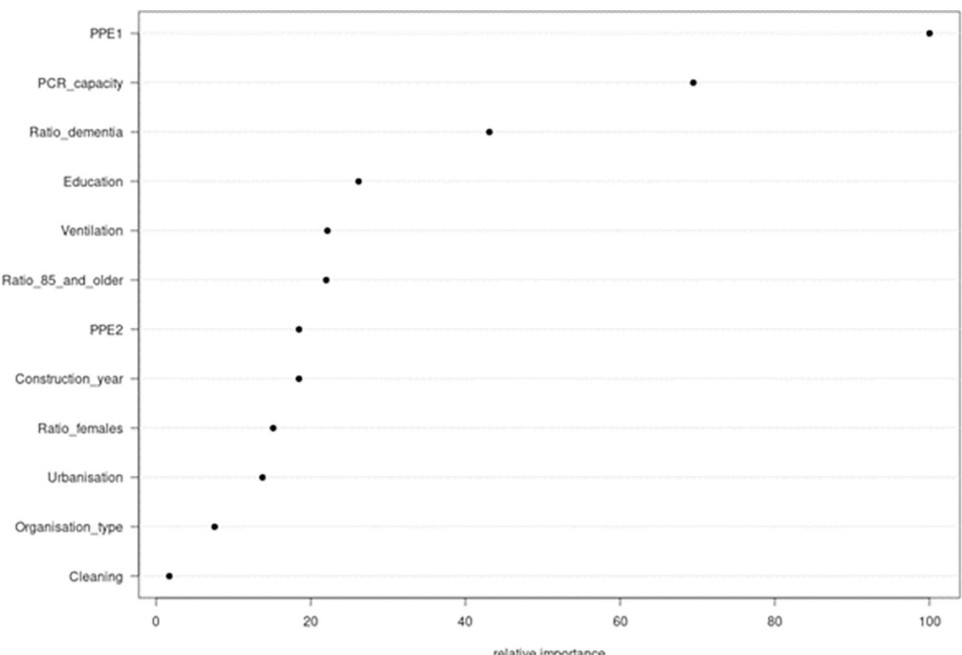

**Fig 2. Graphical presentation of the random forest modeling showing the relative importance of the risk factors in relation to seroprevalence (using IncNodePurity).**

NH may play a role in the spread of SARS-CoV-2 virus within a NH. This exploratory study reports on a number of possible risk factors for higher SARS-CoV-2 seroprevalence in NH.

Due to the limited number of included NH, we applied three methodological approaches to examine the relation between the risk factors and seroprevalence. Overall, consistent results were found for a number of risk factors: the lack of masks (surgical and FFP2), safety glasses and face shields, insufficient PCR-testing capacity organized by government and own resources and a higher percentage of residents with dementia. Ventilation of rooms by opening doors/windows combined with the regular ventilation of common rooms (compared to no such ventilation), and the feeling the staff is enough educated to use the different PPE were correlated with a lower seroprevalence only in the univariable approach. These findings could not be reproduced in the multivariable Bayesian model. Random Forest modeling retained education as the risk factor with the fourth relative importance.

The availability of PPE (surgical and FFP2 masks, safety glasses and face shields) was retained as the most important risk factor in random forest modelling, and was negatively related to seroprevalence in the univariable and multivariable approach. This suggests that shortages of these PPE could have played a major role in the spreading of the virus in NH during the first wave. This is in line with former research, which found that FFP2 mask shortages or delayed FFP2 mask implementation were associated with higher odds for COVID-19 cases, hospitalization and mortality in NH [28–30]. In Belgium, the first case of COVID-19 was reported on February 4[th], 2020, among a passenger who returned from Wuhan, China. From March 18[th], 2020, Belgium applied strict measures of general lockdown. Specifically for the NH, this resulted in banning visitors from March 12[th], 2020. Shortages of masks and testing equipment were quickly becoming apparent, resulting in an advise of the government to reserve them for healthcare personnel and patients at the end of March, 2020. FFP2 masks were reserved for healthcare personnel who had close contact with COVID-19 patients. On social media, calls from healthcare institutions toward companies to bring masks appeared, thus reflecting the shortage.

In our study, a second risk factor was related to the test strategy. In the random forest modelling, this was the second most important factor and the univariable approach showed that NH with insufficient test capacity had significantly higher seroprevalence than NH with enough test capacity (with own resources). Surprisingly, those NH that reported having enough testing capacity with governmental supply and own resources also had higher seroprevalence. Nonetheless, research also shows that outbreak testing, with increased test frequency and use of rapid tests, may be an effective approach to prevent the spread of COVID-19 in NH [31]. But, the efforts put in testing procedures should not hamper the quality of other infection control measures [31]. As mentioned above, not only shortages of PPE, but also testing equipment were a huge challenge during the first wave. On April 8th, 2020, the Flemish health authorities created a task force to prevent the situation in residential care centers from deteriorating further. With respect to testing strategy, only on April 9th, 2020, in Belgium, the test capacity was significantly increased by the government, reserving PCR tests specifically for residential care centers and nursing homes [32]. Despite these efforts, it could not be avoided that in several NH large outbreaks were reported [33]. Our results suggest that these lack of both PPE and testing material may have plaid a role in the spread of the virus in NH.

A positive correlation was found between NH with a higher percentage of residents with dementia and a higher seroprevalence. This may be explained by several reasons. Dementia is a neurodegenerative disorder marked by memory and cognitive problems, behavioral disorders and difficulties performing daily activities, and extensively studied subject in relation to SARS-CoV-2. Research demonstrated that patients with dementia have a higher risk for infection [34] and poorer outcomes when infected, mainly due to their comorbidities [35]. Dementia may further be a risk factor for the spread of the virus, since adherence to sanitary rules and physical distancing measures is very difficult for these patients due to their cognitive state and limited possibilities to cope with changing circumstances [36]. Additionally, a substantial number of the elder people with SARS-CoV-2 infection is asymptomatic or has atypical symptoms [5, 6, 37], which therefore make the diagnosis difficult, especially in those with dementia.

In the univariable approach, ventilation of rooms by opening doors/windows combined with the regular ventilation of common rooms was correlated with a lower seroprevalence, compared to no such a ventilation. Surprisingly, for those who were ventilating rooms by opening rooms and doors but continuously ventilating the commons rooms, no significant relation could be found with seroprevalence. Airborne SARS-CoV-2 transmission has been established as an important transmission route, and several studies showed that ventilation may play a key role in the control of the virus spread [38–40]. Consequently, Heating, Ventilation and Air Conditioning Systems (HVAC) may play an important role in the reduction of transmission. However, when these systems are not used correctly, they may even contribute to the spread of diseases [41]. The latter may explain the absence of a correlation between 'continuous ventilation' and seroprevalence in our findings.

Another factor that was significantly related (only in the univariable model) to lower seroprevalence is the feeling the staff is educated to use the different PPE. This is in line with former research that shows that education and active training may decrease the risk of contamination [42, 43].

## Strengths and limitations

Despite the particular strengths, such as the detailed information on the risk factors and the use of objective seroprevalence (rather than self-reported information on outbreaks) as a measure for the spread of the infection inside a NH, a number of limitations should be mentioned.

First, limitations were related to the premature stop in recruitment due to the start of the second wave of COVID-19. Where only a limited number of NH was available for a large number of risk factors related to seroprevalence to be examined, alternative statistical methods are to be used. While the number of risk factors could indeed be reduced, this resulted in a loss of precision of the information. Also, a number of risk factor variables were treated as continuous variables (PPE and training) to reduce the number of estimates, but are in nature rather ordinal variables. We also have to mention that multiple testing plays a role in the simple model. Especially for those estimates with p-values around 0.05, no conclusions should be made. Therefore, additional analysis applying Bayesian methods was applied to deal with the large number of risk factors for the rather small set of NH. Finally, the upcoming second wave may also have caused a selection in the participating NH: one of the main reasons for refusal to participate was that the NH was confronted with a lot of Covid-19 cases at the time of recruitment. Unfortunately, no comparison can be made in seroprevalence or risk factors between NH which accepted and refused to participate, since we have no questionnaire information nor information about the Covid-19 cases and seroprevalence from these NH. Nevertheless, achieving representativeness is less crucial in analytical studies, were associations are examined [44].

A second important limitation is that we relied on self-reporting for assessing the risk factors: the collective questionnaire was filled in by the director or manager from the NH after the first wave. This implies that responses may be biased by the fact that the respondent is aware of how seriously the NH had been affected by SARS-CoV-2 and by socially desirable answers. The latter is the reason why questions that were explicitly assessing a behavior were not selected as risk factors in the analyses. As a consequence, we want to emphasize that this study should be considered as exploratory.

A last limitation is that we have no information about the number of residents and staff that refused to participate, nor the reasons for refusal, since this information was gathered by the trusted third party.

## Conclusion

This study showed that scarcity of personal protective equipment, inadequate PCR test capacity and prevalence of residents with dementia were related to NH with higher seroprevalence. Our findings suggest that investment and education in infection control are crucial in SARS-CoV-2 control in NH. A more coordinated approach from government is needed to enable a quicker and more adequate reaction on pandemic situations in NH.

## Supporting information

**S1 File. Nursing home specific questionnaire.**
(DOCX)

**S1 Table. Overview of the selected risk factors.** [a]used in unchanged form [b]normalization with the total number of residents [c]normalization with the number of questions [d]combining two variables into one new variable with new factor levels.
(DOCX)

**S1 Fig. Graphical presentation of the random forest modeling showing the relative importance of the risk factors in relation to seroprevalence (using %MSE).**
(DOCX)

## Acknowledgments

The authors wish to thank Lutgart Braeckman, Lieve Neyt, Ann De Muyt, Sylvia Vanden Avenne, Hanne Vercruysse, Ann Herman and Brigitte De Milliano for their contribution in the study protocol, as well as Marc Dhondt, Evi Geyskens, Véronique Coemelck, Isabelle Poiré, Sabrina Chebchoubi, Kyrina Lepere, Lut Hoste and Eveline Nys for their support in the field work and laboratory analysis. Finally, we are very grateful to all participants and the management of the nursing homes.

## Author Contributions

**Conceptualization:** Heidi Janssens, Stefan Heytens, Eline Meyers, Brecht Devleesschauwer, Piet Cools, Tom Geens.

**Data curation:** Heidi Janssens, Tom Geens.

**Formal analysis:** Heidi Janssens, Tom Geens.

**Funding acquisition:** Piet Cools.

**Investigation:** Heidi Janssens, Eline Meyers, Piet Cools, Tom Geens.

**Methodology:** Heidi Janssens, Brecht Devleesschauwer, Piet Cools, Tom Geens.

**Project administration:** Tom Geens.

**Resources:** Heidi Janssens, Piet Cools, Tom Geens.

**Software:** Tom Geens.

**Supervision:** Heidi Janssens, Stefan Heytens, Piet Cools, Tom Geens.

**Validation:** Heidi Janssens, Brecht Devleesschauwer, Piet Cools, Tom Geens.

**Visualization:** Heidi Janssens, Brecht Devleesschauwer, Tom Geens.

**Writing – original draft:** Heidi Janssens, Piet Cools, Tom Geens.

**Writing – review & editing:** Heidi Janssens, Stefan Heytens, Eline Meyers, Brecht Devleesschauwer, Piet Cools, Tom Geens.

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
