## [Decision Letter · Decision Letter 0]

26 Jul 2023

PONE-D-23-12680Exploratory study of risk factors related to SARS-CoV-2 prevalence in nursing homes in Flanders (Belgium) during the first wave of the COVID-19 pandemicPLOS ONE

Dear Dr. Janssens,

Thank you for submitting your manuscript to PLOS ONE. After careful consideration, we feel that it has merit but does not fully meet PLOS ONE’s publication criteria as it currently stands. Therefore, we invite you to submit a revised version of the manuscript that addresses the points raised during the review process.

We look forward to receiving your revised manuscript.

Kind regards,

Amitava Mukherjee, ME, Ph.D.

Academic Editor

PLOS ONE

Journal Requirements:

Reviewers' comments:

Reviewer's Responses to Questions

**Comments to the Author**

1. Is the manuscript technically sound, and do the data support the conclusions?

Reviewer #1: Yes

2. Has the statistical analysis been performed appropriately and rigorously? 

Reviewer #1: Yes

3. Have the authors made all data underlying the findings in their manuscript fully available?

Reviewer #1: Yes

4. Is the manuscript presented in an intelligible fashion and written in standard English?

Reviewer #1: Yes

5. Review Comments to the Author

Reviewer #1: This is a great example of a sound study design and statistical analysis that made use of caution and proper scientific methods to conduct quality research despite unforeseen consequences. The study has been affected by the rapid arrival of Covid-19 wave 2 in the setting, and the anticipated sample size had to be significantly diminished. The authors fully describe this and altered the analysis plan accordingly.

Methods:

1. [lines 105-108] The authors should further describe how the 56 anticipated NH were selected. Was it geographically? How do the 20 that could be included differ from the ones who could not be sampled? What was the acceptance rate? What differences between sampling criteria / available data for those who accepted vs. refused? This should all be reported and discussed in the manuscript to enable the reader to assess selection bias and internal/external validity of the study results.

2. [line 110] What was the number of residents and staff in each of the sampled NH (min, max, mean, SD, median, IQR)? To which proportion of the median do 60 correspond?

3. [line 114] Why were residents living in assisted facilities excluded? What are the potential impacts of this exclusion to the applicability of the results? How was the prevalence and outcomes of Covid-19 different among those in assisted living facilities vs. not?

4. [lines 114-11] What was the refusal rate?

Results:

5. [lines 188-190] The first sentence needs to be clarified. Do the authors mean that 92.2% of the total 1093 individuals included has a valid serological test result? Please rewrite the sentence for clarity. What were the reasons for the absence of a valid serological test in those 7.8%?

6. [lines 210-225] The authors should consider moving this text to the Methods section.

6. PLOS authors have the option to publish the peer review history of their article (what does this mean?). If published, this will include your full peer review and any attached files.

Reviewer #1: **Yes: **Ana Maria Passos-Castilho

---

## [Author Response · Author response to Decision Letter 0]

30 Aug 2023

the responses to the reviewer's questions and comments can be found in the document 'responses to reviewer', which was uploaded separately

---

## [Decision Letter · Decision Letter 1]

26 Sep 2023

Exploratory study of risk factors related to SARS-CoV-2 prevalence in nursing homes in Flanders (Belgium) during the first wave of the COVID-19 pandemic

PONE-D-23-12680R1

Dear Dr. Janssens,

We’re pleased to inform you that your manuscript has been judged scientifically suitable for publication and will be formally accepted for publication once it meets all outstanding technical requirements.

Kind regards,

Amitava Mukherjee, ME, Ph.D.

Academic Editor

PLOS ONE

Additional Editor Comments (optional):

Reviewers' comments:

Reviewer's Responses to Questions

**Comments to the Author**

1. If the authors have adequately addressed your comments raised in a previous round of review and you feel that this manuscript is now acceptable for publication, you may indicate that here to bypass the “Comments to the Author” section, enter your conflict of interest statement in the “Confidential to Editor” section, and submit your "Accept" recommendation.

Reviewer #1: All comments have been addressed

2. Is the manuscript technically sound, and do the data support the conclusions?

Reviewer #1: Yes

3. Has the statistical analysis been performed appropriately and rigorously? 

Reviewer #1: Yes

4. Have the authors made all data underlying the findings in their manuscript fully available?

Reviewer #1: Yes

5. Is the manuscript presented in an intelligible fashion and written in standard English?

Reviewer #1: Yes

6. Review Comments to the Author

Reviewer #1: The authors have satisfactorily addressed all comments, in detail, which have improved the manuscript.

7. PLOS authors have the option to publish the peer review history of their article (what does this mean?). If published, this will include your full peer review and any attached files.

Reviewer #1: **Yes: **Ana Maria Passos-Castilho

---

## [Editor Report · Acceptance letter]

28 Sep 2023

PONE-D-23-12680R1 

Exploratory study of risk factors related to SARS-CoV-2 prevalence in nursing homes in Flanders (Belgium) during the first wave of the COVID-19 pandemic 

Dear Dr. Janssens:

I'm pleased to inform you that your manuscript has been deemed suitable for publication in PLOS ONE. Congratulations! Your manuscript is now with our production department. 

Kind regards, 

on behalf of

Professor Dr. Amitava Mukherjee 

Academic Editor

PLOS ONE